# Transfer NAS with Meta-learned Bayesian Surrogates

**Gresa Shala**[1]**, Thomas Elsken**[2]**, Frank Hutter**[1,2] **& Josif Grabocka**[1]
[1] Department of Computer Science, University of Freiburg
`{shalag,fh,grabocka}@cs.uni-freiburg.de`
[2]Bosch Center for Artificial Intelligence
`thomas.elsken@de.bosch.com`

## Abstract

While neural architecture search (NAS) is an intensely-researched area, approaches typically still suffer from either (i) high computational costs or (ii) lack of robustness across datasets and experiments. Furthermore, most methods start searching for an optimal architecture *from scratch*, ignoring prior knowledge. This is in contrast to the manual design process by researchers and engineers that leverage previous deep learning experiences by, e.g., transferring architectures from previously solved, related problems. We propose to adopt this human design strategy and introduce a novel surrogate for NAS, that is meta-learned across prior architecture evaluations across different datasets. We utilize Bayesian Optimization (BO) with deep-kernel Gaussian Processes, graph neural networks for obtaining architecture embeddings and a transformer-based dataset encoder. As a result, our method consistently achieves state-of-the-art results on six computer vision datasets, while being as fast as one-shot NAS methods.

## 1 Introduction

While deep learning has removed the need for manual feature engineering, it has shifted this manual work to the meta-level, introducing the need for manual architecture engineering. The natural next step is to also remove the need to manually define the architecture. This is the problem tackled by the field of neural architecture search (NAS).

Even though NAS is an intensely-researched area, there is still no NAS method that is both generally robust and efficient. Blackbox optimization methods, such as reinforcement learning (Zoph & Le, 2017), evolutionary algorithms (Real et al., 2019), and Bayesian optimization (Ru et al., 2021; White et al., 2021) work reliably but are slow. On the other hand, one-shot methods (Liu et al., 2019; Dong & Yang, 2019b) often have problems with robustness (Zela et al., 2020), and the newest trend of zero-cost proxies often does not provide more information about an architecture's performance than simple statistics, such as the architecture's number of parameters (White et al., 2022).

An understudied path towards efficiency in NAS is to transfer information across datasets. This idea is naturally motivated by how researchers and engineers tackle new deep learning problems: they leverage the knowledge they obtained from previous experimentation and, e.g., re-use architectures designed for one task and apply or adapt them to a novel task. While a few NAS approaches in this direction exist (Wong et al., 2018; Lian et al., 2020; Elsken et al., 2020; Wistuba, 2021; Lee et al., 2021; Ru et al., 2021), they typically come with one or more of the following limitations: (i) they are only applicable to settings with little data, (ii) they only explore a fairly limited search space or even can just choose from a handful of pre-selected architecture, or (iii) they can not adapt to data seen at test-time. One approach to obtain efficient NAS methods that has been overlooked in the literature so far is to exploit the common formulation of NAS as a hyperparameter optimization (HPO) problem (Bergstra et al., 2013; Domhan et al., 2015; Awad et al., 2021) and draw on the extensive literature on transfer HPO (Wistuba et al., 2016; Feurer et al., 2018a; Perrone & Shen, 2019; Salinas et al., 2020; Wistuba & Grabocka, 2021). In contrast to standard transfer HPO methods that meta-learn parametric surrogates from a pool of source datasets (Wistuba et al., 2016; Feurer et al., 2018a; Wistuba & Grabocka, 2021), in this work we explore the direction of meta learning

surrogates by contextualizing them on the dataset characteristics (a.k.a. meta-features) (Vanschoren, 2018; Jomaa et al., 2021a; Rivolli et al., 2022).

In this work, we present an efficient Bayesian Optimization (BO) method with a novel deep-kernel surrogate that yields a new NAS method which combines the best of both worlds: the reliability of blackbox optimization at a computational cost in the same order of magnitude as one-shot approaches. Concretely, we propose a BO method for NAS that leverages dataset-contextualized surrogates for transfer learning. Following Lee et al. (2021), we use a graph encoder (Zhang et al., 2019) to encode neural architectures and an attention-based dataset encoder (Lee et al., 2019) to obtain context features. We then use *deep kernel learning* (Wilson et al., 2016) to obtain *meta-learned kernels* for the *joint space of architectures and datasets*, allowing us to use the full power of BO for efficient NAS. This approach solves two key issues of Lee et al. (2021), which is closest to our work: (i) a lack of trading-off *exploration vs. exploitation* and (ii) the lack of exploiting new function evaluations on a test task, blindly following what has been observed during meta training. As a result, our surrogates are optimized for efficiently transferring architectures for a new target dataset based on its meta-features. To sum up, our contributions are as follows:

- Inspired by manual architecture design, we treat NAS as a transfer or few-shot learning problem. We leverage ideas from transfer HPO to meta-learn a kernel for Bayesian Optimization, which encodes both architecture and dataset information.
- We are the first to combine deep-kernel Gaussian Processes (GPs) with a graph neural network encoder, a transformer-based dataset encoder, the first to apply BO with deep GPs to NAS, and the first to do all of this in a transfer NAS setting.
- Our resulting method outperforms both state-of-the-art blackbox NAS methods as well as state-of-the-art one-shot methods across six computer vision benchmarks.

To foster reproducibility, we make our code available at `https://github.com/TNAS-DCS/TNAS-DCS`. We address the points in the "NAS Best Practices Checklist" in Appendix F.

## 2 RELATED WORK

NAS is an intensely-researched field, with over 1000 papers published in the last two years alone[1]. We therefore limit our discussion of NAS to the most related fields of Bayesian optimization for NAS and meta learning approaches for NAS. For a full discussion of the NAS literature, we refer the interested readers to a series of surveys by Elsken et al. (2019), Wistuba et al. (2019) and Ren et al. (2020), and for an introduction to BO to Shahriari et al. (2016); Hutter et al. (2019).

**Bayesian optimization (BO) for NAS**. As BO is commonly used in hyperparameter optimization (HPO), one can simply treat architectural choices as categorical hyperparameters and re-use, e.g., tree-based HPO methods that can natively handle categorical choices well (Bergstra et al., 2013; Domhan et al., 2015; Falkner et al., 2018). While Gaussian Processes (GPs) are more typically applied to continuous hyperparameters, they can also be used for NAS by creating an appropriate kernel; such kernels for GP-based BO can be manually engineered (Swersky et al., 2013; Kandasamy et al., 2018; Ru et al., 2021). A recent alternative is to exploit (Bayesian) neural networks for BO (Snoek et al., 2015; Springenberg et al., 2016; White et al., 2021). However, while these neural networks are very expressive, they require more data to fit well than GPs and thus are outperformed by GP-based approaches when only a few function evaluations can be afforded. In this work, we combine the sample efficiency of GPs and the expressive power of neural networks, by using deep GPs combined with a graph neural network encoder.

**Meta learning for NAS.** To mitigate the computational infeasibility of starting NAS methods from scratch for each new task, several approaches have been proposed along the lines of meta and transfer learning. Most of these warm-start the weights of architectures in a target task Wong et al. (2018); Lian et al. (2020); Elsken et al. (2020); Wistuba (2021). Ru et al. (2021) extracts architectural motifs that can be reused on other datasets. Most related to our work is MetaD2A (Lee et al., 2021), where the authors propose to generate candidate architectures and rank them conditioned directly on a task, utilizing a meta-feature extractor (Lee et al., 2019). However, there are two key differences in our

---

[1]See list at: `https://www.automl.org/automl/literature-on-neural-architecture-search`

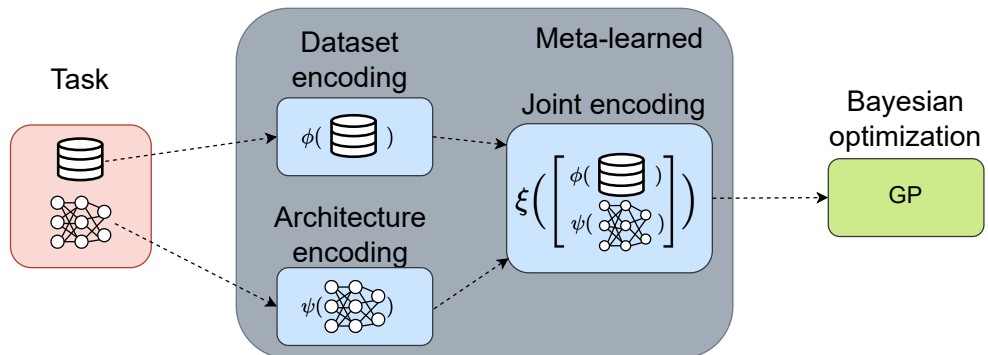

Figure 1: Illustration of TNAS . We employ a GNN $\psi$ to encode architectures, a transformer $\phi$ to encode a dataset, and an MLP $\xi$ to merge the encodings. This joint encoding is then fed into a GP surrogate used within BO. All encodings are meta-learned.

work: (i) the performance predictor of MetaD2A is not probabilistic and thus can not naturally *trade-off exploration vs. exploitation* but rather *only exploits* what has been observed during meta-training. (ii) During the meta testing phase, MetaD2A simply proposes $N$ architectures for a new task, trains the top-$K$ (as estimated by the performance predictor) out of $N$ architectures and returns the best. In particular, the performance of the $K$ evaluated architectures is *not* used as feedback for MetaD2A and thus the method *never* adapts to function evaluations on the test task, blindly following what has been observed during meta training. This can cause problems for new tasks which are poorly correlated with the meta training data. And, indeed, MetaD2A stagnates on several of the datasets in our experimental analysis. In contrast, our approach, dubbed TNAS (transferrable NAS), uses the function evaluations from meta testing to update the surrogate employed within our BO framework, thus allowing to adapt to the meta testing scenario.

**Meta & transfer learning for HPO** There are many approaches to achieve meta or transfer learning in HPO, see, e.g., the survey by Vanschoren (2019) or Feurer et al. (2018b, Section 7). One particularly promising approach is to employ Deep Kernel Learning (Wilson et al., 2016) which strives to learn the kernel function by using a neural network to transform the input to a latent representation, which is then used in a kernel function. Wistuba & Grabocka (2021) and Jomaa et al. (2021b) utilized a deep kernel for transfer learning in HPO. While numerical hyperparameters can be encoded using an MLP, we adapt this approach to NAS by using a graph neural network to encode architectures as inputs; we also extend it by encoding datasets into a latent embedding and learning a deep kernel that spans the combined space of architectures and datasets.

## 3 PROPOSED METHOD

We consider the following problem: given a history of $Q$ datasets, where for each dataset $\mathcal{D}^{(q)}$ we have already evaluated a set of neural network architectures $x_1^{(q)}, \ldots, x_n^{(q)}$ with corresponding performance (e.g., accuracy) $y_1^{(q)}, \ldots, y_n^{(q)}$. For a new dataset $\mathcal{D}^{(\text{new})}$, we want to quickly discover an optimal architecture by leveraging information from the history of datasets. We build upon the state-of-the-art few-shot Bayesian optimization (BO) framework by Wistuba & Grabocka (2021), which was proposed to address a similar problem: transferring optimal hyperparameter configurations across datasets. The authors propose to learn a deep kernel across tasks, which is then used for a Gaussian process (GP) surrogate in the typical BO setup.

However, while hyperparameter configurations can typically be presented by an N-dimensional vector, it is less clear how to represent neural network architectures. Simply representing architectures as vectors and plugging them into an off-the-shelf GP kernel is likely sub-optimal. In fact, White et al. (2020) have shown that the type of architecture representation substantially impacts the performance of a downstream NAS algorithm. To address this issue, we employ graph neural networks (GNNs) to obtain a learnable representation of neural networks. GNNs are a common choice in NAS as neural networks architectures can be naturally represented as graphs (Siems et al., 2020; White et al., 2020; Wen et al., 2020; Dudziak et al., 2020).

Furthermore, while we could directly feed this architecture encoding into the GP's kernel, we argue that the kernel should also be conditioned on the characteristics of a dataset in order to meaningfully asses (dis-)similarities between architectures. To motivate this, consider the question "What makes two architectures similar?". We argue that two architectures are not similar only because they share some similar sub-graph components (which will be represented by the GNN encoding), but also because they achieve similar performance on the target dataset. Following this line of reasoning, we condition the deep kernel on the characteristics (meta-features) of a dataset (Vanschoren, 2018). In a similar fashion as for the architecture encoding, we again use a learnable representation of datasets via employing a set transformer, as also done by (Lee et al., 2019). The architecture and dataset encoding are then processed by a fully-connected neural network, whose output serves as the input for an off-the-shelf kernel function, e.g., a Matérn kernel, which is finally used to compute the distance of two (architecture, dataset) datapoints. This results in an end-to-end learnable encoding of the problem, and the parameters of the GNN, transformer and fully-connected neural network are meta-learned in a similar fashion as in Wistuba & Grabocka (2021). We refer to Figure 1 for an overview of our framework.

In the following subsection, we discuss the details of the different components.

## 3.1 BAYESIAN OPTIMIZATION WITH DEEP KERNEL GAUSSIAN PROCESSES

We start by introducing Gaussian Processes, which represents the surrogate of our method within Bayesian optimization (BO). In a typical hyperparameter optimization (HPO) setup, the inputs $x \in \mathcal{X}$ represent hyperparameter configurations, and the target $y \in \mathcal{Y}$ denotes the performance of a machine learning method when trained with the hyperparameter configuration $x$. Consider the training $\mathcal{D} = \{(x_i, y_i)\}_{i=1}^{n}$ and testing $\mathcal{D}^* = \{(x_i^*, y_i^*)\}_{i=1}^{n^*}$ splits of a dataset of evaluated hyperparameters. In that context, GPs are non-parametric models that assume a prior over functions, and approximate the target $y \in \mathcal{Y} \subseteq \mathbb{R}_+$ given the features $x \in \mathcal{X} \subseteq \mathbb{R}^L$. The estimation of the target variable $y^*$ for the test instances $x^*$ is also jointly Gaussian as $\begin{bmatrix} y \\ y^* \end{bmatrix} \sim \mathcal{N}\left(0, \begin{pmatrix} K(x,x) & K(x,x^*) \\ K(x,x^*)^T & K(x^*,x^*) \end{pmatrix}\right)$. Each respective block of the covariance matrix is the result of applying a kernel function $k : \mathcal{X} \times \mathcal{X} \to \mathbb{R}_+$ on pairs of instances, e.g., $K(x,x^*)_{i,j} := k(x_i, x_j^*)$. The estimated posterior mean and covariance of GPs (Rasmussen & Williams, 2006) for the target $y^*$ of the test instances $x^*$ is given as follows:

$$\mathbb{E}[y^* \mid x^*, x, y] = K(x^*, x)K(x,x)^{-1}y \tag{1}$$

$$\mathrm{cov}[y^* \mid x^*, x] = K(x^*, x^*) - K(x,x^*)^T K(x,x)^{-1} K(x,x^*). \tag{2}$$

We refer to, e.g., Murphy (2012) for the derivation. GPs are *lazy* models that rely on the similarity of the test instances to the training instances via kernel functions $k$, such as the Matérn kernel. Unfortunately, typical kernels used with GPs are designed manually and rely on sub-optimal assumptions (Cowen-Rivers et al., 2020), which deteriorates the GP's performance. A promising direction for designing powerful and efficient kernel functions that adapt to a learning task is Deep Kernel Learning (Wilson et al., 2016), where kernels are represented as trainable neural networks. A mapping $\xi : \mathcal{X} \to \mathbb{R}^L$ projects the features to a latent representation, where similar instances are co-located.

**The embedding $\xi$ for the deep kernel** of our GPs is a fully-connected neural network, that takes as input the encoding of the architecture $\psi$ as well as the dataset encoding $\phi$. In detail, the $L$-dimensional architecture encoding $\psi$ is fused with the $K$-dimensional dataset encoding $\phi$ and processed through a fully connected neural network $\xi : \mathbb{R}^{K+L} \to \mathbb{R}^M$, where the last layer has $M$ neurons. We re-use both the GNN-based architecture encoding and the transformer-based dataset encoding from Lee et al. (2021), thus we only briefly describe it below and refer to Lee et al. (2021) for details.

**The architecture encoding $\psi$** consists of a directed acyclic *graph encoder* (Zhang et al., 2019) to obtain the encoding for the architectures. By using one GRU cell to traverse the topological order of the DAG in the direction from the input to the output, and another GRU cell to pass through the DAG in the backward direction, we obtain latent representations of the graph which are then put through a fully-connected neural network layer to obtain the encoding for the architecture.

**The dataset encoding $\phi$** consists of two stacked *Set-Transformer* (Lee et al., 2019) architectures. The first Set-Transformer layer captures the interaction between randomly sampled data points of the same class, whereas the second one captures the interactions between the different classes of the dataset. The resulting output of the second Set-Transformer layer represents the dataset encoding.

We tuned the dimensionality of the embedding of the dataset encoder and graph encoder, as well as the architecture of the feed-forward neural network of our method using the multi-fidelity Bayesian optimization method BOHB (Falkner et al., 2018) on the meta-training dataset; please refer to Appendix A for details.

**Putting it all together,** the kernel/similarity between two architectures, specifically $\mathbf{x}$ evaluated on dataset $\mathcal{D}$, and $\mathbf{x}'$ evaluated on dataset $\mathcal{D}'$, is:

$$k\left(\mathbf{x}, \mathcal{D}, \mathbf{x}', \mathcal{D}'; w\right) = k\left( \xi\left( \left[ \psi(\mathbf{x}; w^{(\psi)}), \phi\left(\mathcal{D}; w^{(\phi)}\right) \right]; w^{(\xi)} \right), \right.$$

$$\left. \xi\left( \left[ \psi(\mathbf{x}'; w^{(\psi)}), \phi\left(\mathcal{D}'; w^{(\phi)}\right) \right]; w^{(\xi)} \right); w^{(k)} \right)$$

with $w^{(\xi)}$ being the parameters of the neural network $\xi$, $w^{(\psi)}$ the parameters of the architecture encoding $\psi$, $w^{(\phi)}$ the parameters of the dataset encoding $\phi$, and $w^{(k)}$ additional parameters of the kernel function. We denote the cumulative parameters as $w := \left(w^{(\xi)}, w^{(\psi)}, w^{(\phi)}, w^{(k)}\right)$. All parameters are jointly meta-learned to maximize the marginal likelihood (Wistuba & Grabocka, 2021), as will be discussed in the next section.

### 3.2 META-LEARNING DEEP-KERNEL GP SURROGATES

Recall that we assume we are given a set of $Q$ datasets, where on each dataset $\mathcal{D}_q$ we have $N_q \in \mathbb{N}_+$ evaluated architectures. We denote the $n$-th architecture evaluated on the $q$-th dataset as $x_{q,n}$ and its validation accuracy as $y_{q,n}$. The meta-dataset of all the evaluations on all the datasets is defined as $\mathcal{M} := \bigcup_{q=1}^{Q} \bigcup_{n=1}^{N_q} \{(x_{q,n}, y_{q,n}, \mathcal{D}_q)\}$. By $\mathbf{x} := (x_{1,1}, \ldots, x_{Q,N_q}), \mathbf{y} := (y_{1,1}, \ldots, y_{Q,N_q})$ and $\mathcal{D} := (\mathcal{D}_1, \ldots, \mathcal{D}_Q)$ we denote vectors containing all the architectures, accuracies and datasets.

The parameters $w$ of the deep kernel are optimized jointly by maximizing the log marginal likelihood of the GP surrogate on the meta training dataset:

$$\arg\max_{w} \ \log p\left(\mathbf{y} \mid \mathbf{x}, \mathcal{D} \, ; \, w\right) \tag{3}$$

$$\propto \arg\min_{w} \ \mathbf{y}^T K^{-1}(\mathbf{x}, \mathcal{D}; w) \, \mathbf{y} + \log |K(\mathbf{x}, \mathcal{D}; w)|.$$

In practice, we resort to sampling mini-batches and employ stochastic gradient descent, following established practices (Wilson et al., 2016; Wistuba & Grabocka, 2021; Patacchiola et al., 2020). As all components of our method are differentiable, our approach is end-to-end differentiable.

For updating the meta-parameters $w$, we use the meta-learning algorithm REPTILE (Nichol et al., 2018), due to its simplicity compared to, e.g.,

---

**Algorithm 1** Meta-learning our deep-kernel GPs

1: **Require:** meta-dataset $\mathcal{M}$; learning rates $\eta_{SGD}, \eta_{REP}$; inner update steps $v$.
2: **while** not converged **do**
3:   Sample mini-batch from $\mathcal{M}$:
    $\mathbf{x} = [x_1, \ldots, x_k], \mathbf{y} = [y_1, \ldots, y_k], \mathcal{D} = [\mathcal{D}_1, \ldots, \mathcal{D}_k]$
4:   $\mathcal{L}(w) = \mathbf{y}^T K^{-1}(\mathbf{x}, \mathcal{D}; w)\mathbf{y} + \log |K(\mathbf{x}, \mathcal{D}; w)|$
5:   $w' \leftarrow w$
6:   **for** $j = 1$ to $v$ **do**
7:     $w' \leftarrow w' - \eta_{SGD} \nabla_{w'} \mathcal{L}(w')$
8:   Update $w \leftarrow w - \eta_{REP}(w - w')$

---

MAML (Finn et al., 2017). Our method's pseudocode is shown in Algorithm 1. The procedure samples a mini-batch of (architecture, dataset) pairs with corresponding validation accuracy (line 3). We sample datasets and architectures uniformly at random from the meta-training dataset. Then we fit the surrogate to estimate the accuracies $y$ of architectures $x$ on a dataset $\mathcal{D}$ using SGD by minimizing Equation 3 (lines 4-7). Finally, we use REPTILE (Nichol et al., 2018) to update $w$ (line 8).

### 3.3 META-TESTING

Once the optimal $w$ are found, we plug in the meta-learned kernel for a GP, with the posterior and use vanilla Bayesian Optimization (BO) to quickly identify the optimal configuration in the new response surface. Note that by employing BO in the meta-testing phase, TNAS *adapts to the test task* by updating the posterior of the GP surrogate with function evaluations from the test task (remember equations 1, 2). We offer a more detailed description of the BO loop in Appendix C.

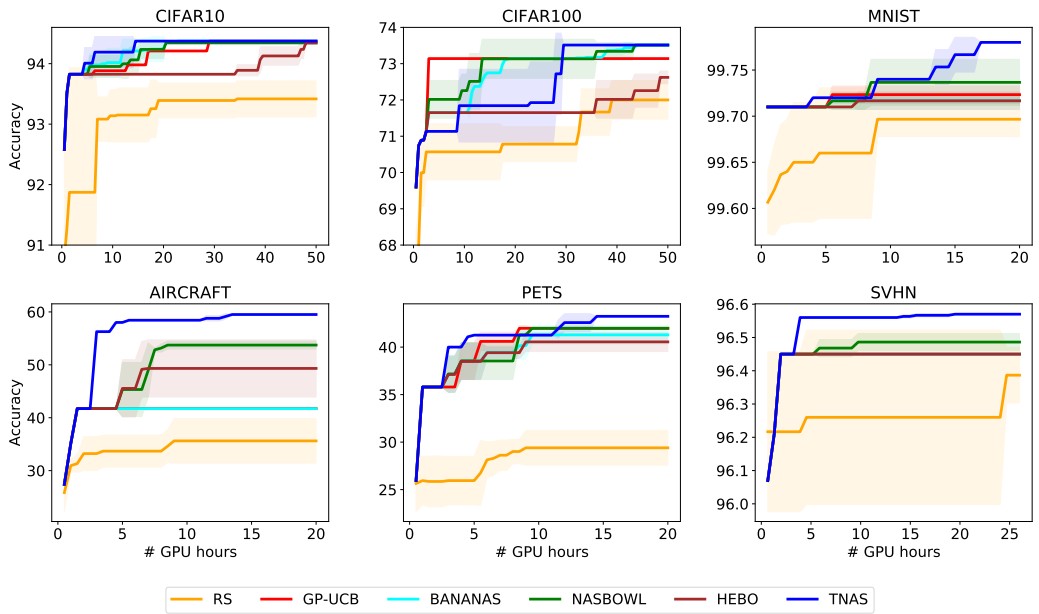

Figure 2: Comparing TNAS to random search (RS) and the four different Bayesian optimization methods on six image datasets.

# 4 EXPERIMENTAL SETUP

We follow the experimental setup as Lee et al. (2021) for the NAS-Bench-201 (Dong & Yang, 2020) and MobileNetV3 search spaces. On the NAS-Bench-201 search space, we also use the same meta datasets as Lee et al. (2021). It consists of 4230 meta-training datasets derived from ImageNet. For each of these datasets, the accuracy of one (different) architecture from the NAS-Bench-201 search space is given. For the evaluation of our method and the baselines, we use six popular computer vision datasets: CIFAR-10, CIFAR-100, SVHN, Aircraft, Oxford IIT Pets, and MNIST. For CIFAR-10 and CIFAR-100, we query the performances of architectures from the NAS-Bench-201 benchmark, whereas for the other four datasets we train the suggested architectures from scratch using the NAS-Bench-201 pipeline (as these are not available in the benchmark). We ran three trials for each experiment and report the mean and standard deviations.

## 4.1 BASELINES

**Classic HPO.** A simple baseline is Random Search (RS) (Bergstra & Bengio, 2012). RS samples architectures uniformly at random from the search space and returns the top preforming one. Another simple yet powerful baseline is Bayesian Optimization with a vanilla GP surrogate (Snoek et al., 2012). We use the Matérn $5/2$ kernel and rely on the GPytorch (Gardner et al., 2018) implementation. We tried both Expected Improvement (EI) and Upper Confidence Bound (UCB) as acquisition functions, with UCB performing better in our experiments. We also compare to HEBO (Cowen-Rivers et al., 2020), a black-box HPO method that performs input and output warping to mitigate the effects of heteroscedasticity and non-stationarity on HPO problems. HEBO won the 2020 NeurIPS blackbox optimization challenge (Turner et al., 2021). We use the implementation provided by the authors.

**HPO for NAS.** White et al. (2021) proposed a BO method for NAS that uses an ensemble of fully-connected neural networks as a surrogate, named BANANAS. Moreover, BANANAS uses a path encoding for the neural architectures, which serves as an input to the ensemble. When applied to a new test task, BANANAS starts the neural architecture search from scratch. NASBOWL (Ru et al., 2021) is a GP-based BO method for NAS and utilizes the Weisfeiler-Lehman kernel (Shervashidze et al., 2011). We use the implementations provided by the authors for BANANAS and NASBOWL.

**State-of-the-art in NAS.** One-shot methods have recently shown strong empirical performance for NAS. We compare to GDAS (Dong & Yang, 2019b), SETN (Dong & Yang, 2019a), PC-DARTS (Xu

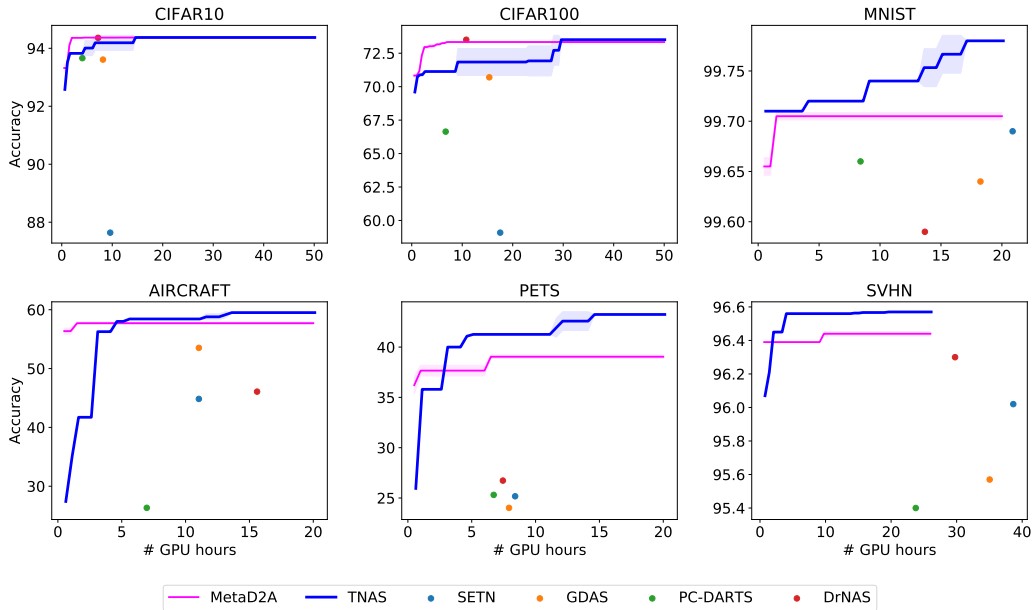

Figure 3: Comparison of TNAS to state-of-the-art NAS methods.

et al., 2020) and DrNAS (Chen et al., 2021). For these methods, we compare to published results from the literature for six computer vision datasets.

**Transfer NAS.** We compare to the most related transfer NAS method, MetaD2A (Lee et al., 2021), which is a transfer learning NAS method with a dataset-contextualized neural network generator and performance predictor. The neural network generator and performance predictor are meta-trained on the same source datasets that we also use. When applied to a test task, MetaD2A generates 500 candidate architectures conditioned on the test dataset and then selects the top architectures based on its performance predictor. We use the implementation provided by MetaD2A's authors.

## 5 RESEARCH HYPOTHESES AND EXPERIMENTAL RESULTS

Our experiments are designed to validate the following research hypotheses for our approach, dubbed TNAS :

**Hypothesis 1**: *TNAS is more efficient than classical HPO methods applied to NAS as well as HPO methods specifically adapted to NAS and outperforms them in terms of anytime-performance, while achieving strong final performance.*

**Hypothesis 2**: *TNAS is competitive with one-shot approaches in terms of runtime.*

In summary, we validate our claim from the introduction:

**Hypothesis 3**: *TNAS achieves the consistency of blackbox optimization algorithms (such as classical HPO methods) while being as efficient as one-shot methods.*

**Results for Hypothesis 1.**   In Figure 2, we compare the performance of TNAS, with several HPO baselines. For all methods, we use the 5 top-performing architectures from the meta-training dataset as a starting point. In that sense, all these baselines are "transfer learning" by being initialized with the best architectures on the meta-training dataset. On all of the datasets except CIFAR100, TNAS finds top-performing architectures faster than all the baselines and achieves stronger anytime performance. Furthermore, on all benchmarks, TNAS eventually performs best.

**Results for Hypothesis 2.**   We demonstrate that with our meta-learned deep kernel within Bayesian Optimization, the search time can be significantly reduced, to the same order as one-shot approaches. Figure 3 shows the performance of both TNAS and MetaD2A compared to state-of-the-art NAS

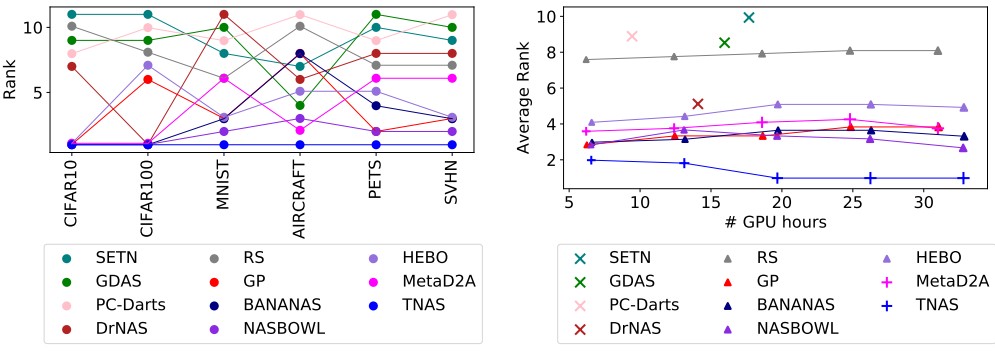

(a) Ranking of TNAS across benchmarks.

(b) Ranking (averaged across benchmarks) over the course of runtime.

Figure 4: Consistency of TNAS compared to baselines.

methods. Except for CIFAR10 and CIFAR100, TNAS clearly outperforms the NAS baselines by the time the one-shot approaches finish the search. On CIFAR10, TNAS achieves similar performance as the baselines, while it is slightly inferior on CIFAR100 for some baselines and only overtakes them given more time. We refer to Table 3 in the appendix for concrete numbers. We can furthermore observe the drawbacks of MetaD2A discussed earlier: (i) MetaD2A does not *trade-off exploration vs. exploitation* but rather *only exploits* what has been observed during meta-training and (ii) MetaD2A does not use evaluations on the new target dataset as feedback and *never* adapts. As a result, MetaD2A stagnates on several of the datasets.

**Results for Hypothesis 3.** In Figure 4, we show that TNAS consistently achieves strong results, while the existing state-of-the-art NAS baselines have much higher variance - their ranking changes across benchmarks. Furthermore, the figure shows how the ranking evolves over the course of running of the methods (by means of runtime in GPU hours). The analysis indicates that TNAS consistently achieves the best performance, for both small and large computational budgets, in particular also when compared to MetaD2A.

## 5.1 ABLATING OUR DESIGN CHOICES

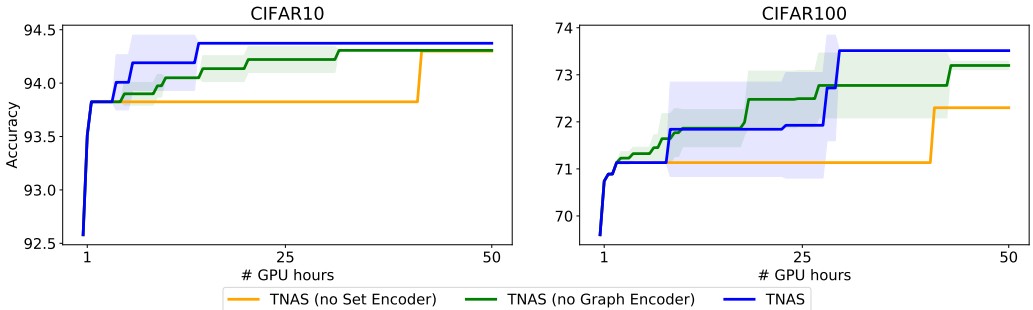

Figure 5: Ablation of the components of TNAS .

We empirically analyse our design choices, namely the graph and dataset encoder and demonstrate a lift in performance compared to *ceteris paribus* ablations that do not employ these designs.

Concretely, we ablate our method's components and test them on CIFAR-10 and CIFAR-100, reporting the result in Figure 5. *TNAS (no Set Encoder)* shows the performance of our method using a graph encoder, but no set-encoder (i.e., without using any dataset meta-features). *TNAS (no Graph Encoder)* shows the performance of our method using the dataset encoding in combination with a matrix encoding for the architectures (i.e., no graph encoder). *TNAS* outperforms the other variations; thus, we conclude that *using both the learnable dataset meta-features and a graph neural network* encoding is the most robust surrogate design. This finding validates the design choices of our method.

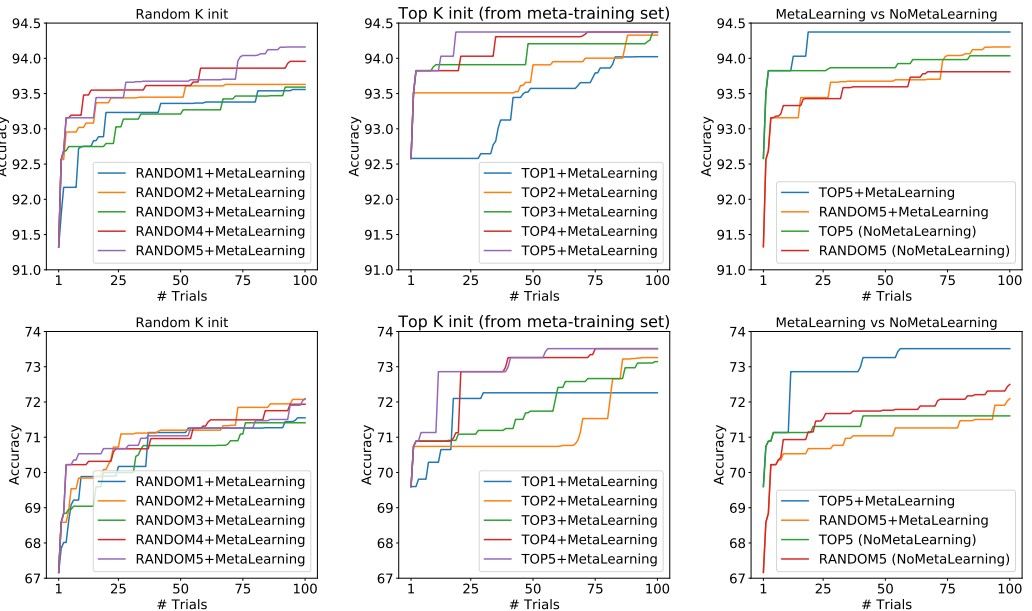

Figure 6: Ablation of the initial design and the initialization of our method on CIFAR-10 (top row) and CIFAR-100 (bottom row). Left: using up to 5 random architectures; middle: using up to 5 top architectures from the meta-training set; right: ablating the comparison of (i) a meta-learned surrogate and a randomly initialized one with (ii) a random initial design and an initial design from the top 5 architectures from the meta-training set.

We also empirically evaluated (i) whether there is actually a benefit in meta-learning the surrogate, and (ii) our method's performance with different initial architectures. Remember that as an initial design for our method, we used the top-5 performing architectures from the meta-training dataset. As alternatives, we consider values other than 5 and also start from randomly sampled architectures. We also turn-off meta-learning. The plots for these experiments are shown in Figure 6, evaluated on CIFAR-10 and CIFAR-100. The results suggest that both design choices are beneficial.

# 6 CONCLUSIONS AND FUTURE WORK

NAS is an intensely researched task and in essence is an instance of the hyperparameter optimization (HPO) problem. In this work, we exploited this relationship and, motivated by state-of-the-art transfer HPO methods, adapted deep Gaussian Process (GP) surrogates to capture architecture representations. Furthermore, we proposed a novel conditioning of the deep GP on dataset meta-features to enable transferring well-performing architectures from source datasets with similar meta-features. We empirically motivated the impact of each component of our proposed method through extensive ablations. In addition, we showed that our novel deep GPs with dataset meta-features and architecture encodings achieve the highest accuracy on six computer vision datasets compared to a broad range of HPO methods, BO methods for NAS, and one-shot NAS methods. Lastly, we demonstrated that proxy architecture evaluations allow our method to discover more accurate architectures within the same search time one-shot NAS methods require.

**Future work.** A deep surrogate in principle allows us to capture the interaction between architectures and hyperparameter configurations. The NAS literature at the moment underexplores the impact of hyperparameters on the performance of an architecture. In fact, it is commonly known that the same architecture would perform differently if the training pipeline is altered, for example by changing the learning rate, the number of epochs, or the degree of regularization. Our previously-defined surrogate can be trivially extended to model the interaction of architecture embeddings, the dataset meta-features and hyperparameter configurations. We have not explored this direction empirically due to the lack of available NAS meta-datasets that vary both architectures and hyperparameters but would like to do in the future.

ACKNOWLEDGEMENTS

We acknowledge Hadi Jomaa as a contributor to this paper on the level of a co-author. We would like to acknowledge the grant awarded by the Eva-Mayr-Stihl Stiftung. In addition, this research was funded by the Deutsche Forschungsgemeinschaft (DFG, German Research Foundation) under grant number 417962828. In addition, Josif Grabocka acknowledges the support of the BrainLinks-BrainTools center of excellence.

**Reproducibility Statement.** To foster reproducibility, we make our code available at `https://github.com/TNAS-DCS/TNAS-DCS`. We give details on our experimental protocol in the "NAS Best Practices Checklist" in Appendix F.

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

## A   HYPERPARAMETER OPTIMIZATION FOR GNN AND SET TRANSFORMER ARCHITECTURES

We tuned the dimensionality of the embedding of the dataset encoder and graph encoder (*Embedding dims.*), the architecture of the feed-forward neural network of our method (*Num. layers*, *Num. units in layer 1*, *Num. units in layer 2*, *Num. units in layer 3*, *Num. units in layer 4*), and the learning rate of the joint meta-training using BOHB (Falkner et al., 2018) on the meta-training dataset.

| Hyperparameter | Range |
|---|---|
| Embedding dims. | $\{8, 16, 20, 28, 32, 48, 56\}$ |
| Num. layers | $[1, 4]$ (integer) |
| Num. units in layer 1 | $[16, 512]$ (log space) |
| Num. units in layer 2 | $[16, 512]$ (log space) |
| Num. units in layer 3 | $[16, 512]$ (log space) |
| Num. units in layer 4 | $[16, 512]$ (log space) |
| Learning rate | $[10^{-6}, 10^{-1}]$ (log space) |

Table 1: The hyperparameter space for tuning our method with BOHB (Falkner et al., 2018).

## B   ARCHITECTURE ENCODING IN TNAS

To encode the architectures, we use a directed acyclic *graph encoder* Zhang et al. (2019). It takes architectures represented as a graph as input, and provides as output vector representations for those architectures. Following Zhang et al. (2019), the graph encoder consists of two GRU cells and a fully-connected neural network layer. Each GRU cell traverses the graph representation of the architecture in opposite directions to the other. The output of both cells is then concatenated and input through the fully-connected layer, which outputs the vector representation that will be used as an encoding for the architecture. We meta-train the graph encoder, as well as the dataset encoder, jointly with the embedding of the deep kernel during meta-training to maximize the log marginal likelihood of the GP surrogate.

## C   BAYESIAN OPTIMIZATION WITH TNAS

Once we meta-train our surrogate on the meta-training set using Algorithm 1, we use it at meta-testing time in a Bayesian optimization (BO) loop. First, we evaluate the top-5 architectures from the meta-training set on the test dataset. Once the GP is fit to these architectures and their respective accuracies, we use EI as the acquisition function to find the next architecture to evaluate. In the NASBench201 search space, we repeat this BO loop for a total of 100 evaluations for CIFAR10 and CIFAR100, whereas for SVHN, Aircraft, Pets, and MNIST, we do 40 evaluations. In the MobilenetV3 search space we do 50 evaluations for each of the datasets.

For maximizing the acquisition function in order to find the next candidate architecture, we either exhaustively evaluate the entire search space (in the case of small search spaces such as NAS-Bench-201), or simply sample K architectures randomly and pick the best (in the case of the MobileNet V3 space, $K = 10^4$).

## D   COMPARISON OF TNAS AND METAD2A

We refer to Table 2 for a comparison of the key ingredients of Lee et al. (2021) and our approach.

| | predictor | candidate architecture generation | test-time adaptation |
|---|---|---|---|
| Lee et al. (2021) | (dataset, architecture) encoding, MLP (deterministic prediction) | GNN-based architecture generator, select top-K based on predictor | no |
| Ours | (dataset, architecture) encoding, MLP, GP (probabilistic prediction) | BO, maximize acquisition function | yes, adapt GP predictor w.r.t. new candidate evaluations |

Table 2: Comparison of our work to Lee et al. (2021).

## E    COMPARISON OF TNAS TO SOTA NAS METHODS AND MOBILENET V3 SPACE

Table 3: Accuracy of our method (TNAS ) in terms of time and accuracy compared to state-of-the-art NAS methods on the NASBench201 search space.

| Data | Method | GPU days | Accuracy |
|---|---|---|---|
| CIFAR-10 | SETN | 0.40 | $87.64_{\pm 0.00}$ |
| | GDAS | 0.34 | $93.61_{\pm 0.09}$ |
| | PC-DARTS | 0.17 | $93.66_{\pm 0.17}$ |
| | DrNAS | 0.30 | $94.36_{\pm 0.00}$ |
| | arch2vec | 1.38 | $91.41_{\pm 0.22}$ |
| | MetaD2A | 2.08 | $\mathbf{94.37}_{\pm 0.00}$ |
| | **TNAS** | 2.18 | $\mathbf{94.37}_{\pm 0.00}$ |
| CIFAR-100 | SETN | 0.73 | $59.09_{\pm 0.24}$ |
| | GDAS | 0.64 | $70.70_{\pm 0.30}$ |
| | PC-DARTS | 0.28 | $66.64_{\pm 2.34}$ |
| | DrNAS | 0.45 | $\mathbf{73.51}_{\pm 0.00}$ |
| | arch2vec | 1.38 | $73.35_{\pm 0.32}$ |
| | MetaD2A | 2.08 | $\mathbf{73.51}_{\pm 0.15}$ |
| | **TNAS** | 2.18 | $\mathbf{73.51}_{\pm 0.00}$ |
| MNIST | SETN | 0.87 | $99.69_{\pm 0.04}$ |
| | GDAS | 0.76 | $99.64_{\pm 0.04}$ |
| | PC-DARTS | 0.35 | $99.66_{\pm 0.04}$ |
| | DrNAS | 0.57 | $99.59_{\pm 0.02}$ |
| | MetaD2A | 0.83 | $99.71_{\pm 0.02}$ |
| | **TNAS** | 0.89 | $99.78_{\pm 0.00}$ |
| Aircraft | SETN | 0.46 | $44.84_{\pm 3.96}$ |
| | GDAS | 0.46 | $53.52_{\pm 0.48}$ |
| | PC-DARTS | 0.29 | $26.33_{\pm 3.40}$ |
| | DrNAS | 0.65 | $46.08_{\pm 7.00}$ |
| | MetaD2A | 0.83 | $57.71_{\pm 0.72}$ |
| | **TNAS** | 0.89 | $\mathbf{59.51}_{\pm 0.0}$ |
| Pets | SETN | 0.35 | $25.17_{\pm 1.68}$ |
| | GDAS | 0.33 | $24.02_{\pm 2.75}$ |
| | PC-DARTS | 0.28 | $25.31_{\pm 1.38}$ |
| | DrNAS | 0.31 | $26.73_{\pm 2.61}$ |
| | MetaD2A | 0.83 | $39.04_{\pm 0.72}$ |
| | **TNAS** | 0.89 | $\mathbf{43.24}_{\pm 0.0}$ |
| SVHN | SETN | 1.61 | $96.02_{\pm 0.04}$ |
| | GDAS | 1.46 | $95.57_{\pm 0.04}$ |
| | PC-DARTS | 0.99 | $95.40_{\pm 0.04}$ |
| | DrNAS | 1.24 | $96.30_{\pm 0.02}$ |
| | MetaD2A | 1.08 | $96.44_{\pm 0.05}$ |
| | **TNAS** | 1.18 | $\mathbf{96.57}_{\pm 0.00}$ |

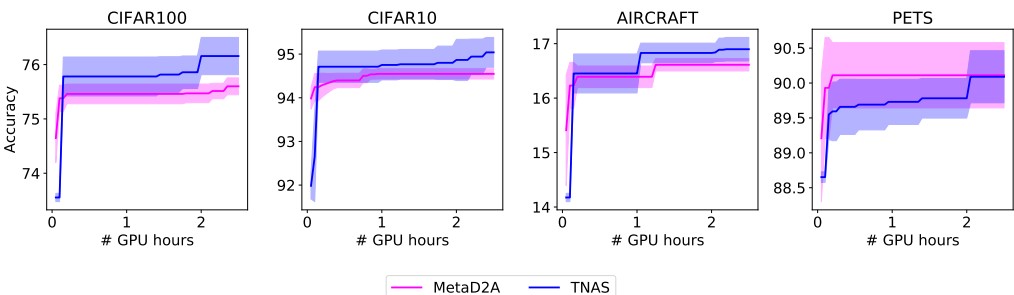

Figure 7: Comparison of TNAS to MetaD2A on the MobileNetV3 search space.

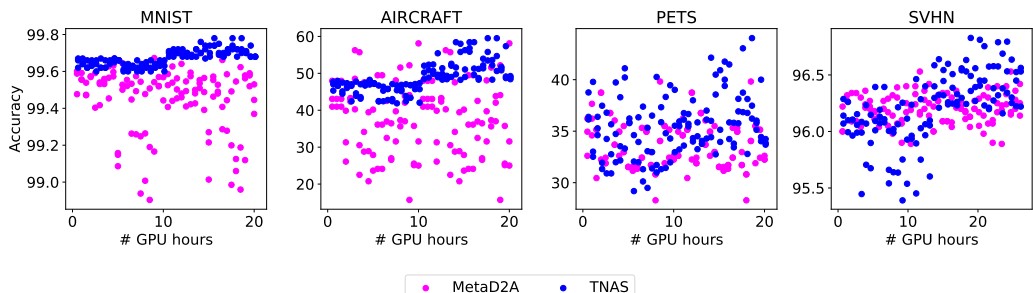

Figure 8: Comparison of TNAS to MetaD2A in terms of the accuracy of the suggested architectures to evaluate on the NASBench201 search space. For MetaD2A the architectures generated by its architecture generator are evaluated sequentially based on MetaD2A's predictor ranking. TNAS adapts during the BO loop iterations, and thus suggests architectures conditioned on the previous evaluations.

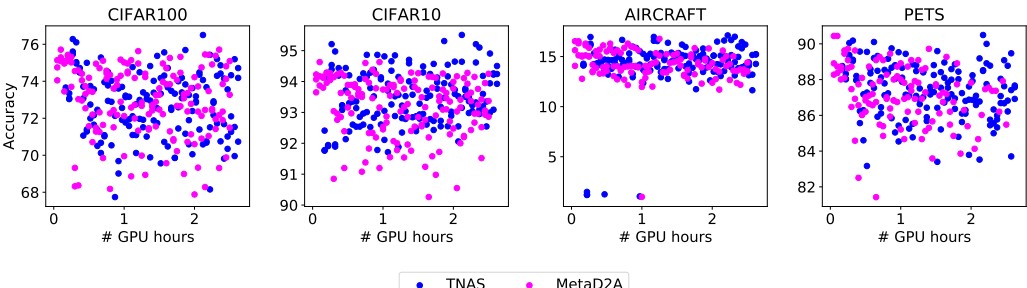

Figure 9: Comparison of TNAS to MetaD2A in terms of the accuracy of the suggested architectures to evaluate on the MobileNetV3 search space.

## F  NAS BEST PRACTICE CHECKLIST

We now describe how we addressed the individual points of the NAS best practice checklist (Lindauer & Hutter, 2020).

1. **Best Practices for Releasing Code**

   For all experiments you report:
   (a) Did you release code for the training pipeline used to evaluate the final architectures? The code for the training pipeline for the architectures can be found in the repo we provide.
   (b) Did you release code for the search space? We used the NAS-Bench-201 search space in our experiments, the description and code for which is publicly available.
   (c) Did you release the hyperparameters used for the final evaluation pipeline, as well as random seeds? We the NAS-Bench-201 pipeline and hyperparameters as our final evaluation pipeline. We release it as well as the random seeds in the repo we provide.
   (d) Did you release code for your NAS method? The code for our NAS method can be found in `https://anonymous.4open.science/r/TNAS-DCS-CC08`.
   (e) Did you release hyperparameters for your NAS method, as well as random seeds? The hyperparameters for our NAS method, as well as random seeds for the experiments can be found in the repo we provide.

2. **Best practices for comparing NAS methods**
   (a) For all NAS methods you compare, did you use exactly the same NAS benchmark, including the same dataset (with the same training-test split), search space and code for training the architectures and hyperparameters for that code?? Yes, for fair comparison we made sure to use the same evaluation pipeline for all NAS methods we compare.
   (b) Did you control for confounding factors (different hardware, versions of DL libraries, different runtimes for the different methods)? We ran all the methods on the same hardware and the same environment to control for confounding factors.
   (c) Did you run ablation studies? We ran extensive ablation studies on the components of our method.
   (d) Did you use the same evaluation protocol for the methods being compared? Yes.
   (e) Did you compare performance over time? Yes.
   (f) Did you compare to random search? Performance comparison to random search as well as other baselines can be found in Figure 2.
   (g) Did you perform multiple runs of your experiments and report seeds? For each of the experiments we performed three runs with different seeds ($333, 444, 555$).
   (h) Did you use tabular or surrogate benchmarks for in-depth evaluations? We used NAS-Bench-201 as a tabular benchmark.

3. **Best practices for reporting important details**
   (a) Did you report how you tuned hyperparameters, and what time and resources this required? We use BOHBFalkner et al. (2018) to tune the hyperparameters of our method. We ran BOHB with three different random seeds for $24$ hours.
   (b) Did you report the time for the entire end-to-end NAS method (rather than, e.g., only for the search phase)? Yes.
   (c) Did you report all the details of your experimental setup? The details of our experimental setup can be found in Section 4.

