# OpenReview forum: "Transfer NAS with Meta-learned Bayesian Surrogates"
_ICLR.cc/2023/Conference — ICLR 2023 notable top 5%_

### Official Review · Reviewer_3f7e · 2022-10-18

**Confidence:** 4
**Correctness:** 4
**Technical Novelty And Significance:** 3
**Empirical Novelty And Significance:** 3
**Recommendation:** 8

**Clarity, Quality, Novelty And Reproducibility:**

### Quality
The writing and organization of this paper are great.

### Clarity
All contents are clear and easy to follow.

### Novelty
It's novel to leverage the previous knowledge in NAS, although each component, i.e., learning architecture and dataset embedding, BO are not new.

### Reproducibility
Code is provided, and it provides a NAS Best Practices Checklist.




**Strength And Weaknesses:**

### Strengths
- The motivation of this paper is great, the idea of transfer previous knowledge is straightforward and lack of exploration. To my knowledge, there are previous methods trying to learn from history [1], but using on NAS is novel.
- The results on NAS-Bench-201 are pretty robust across six datasets, and it's good to see from the ablation study that both the architecture and the dataset embedding have contributions to the final performance.

### Weaknesses
- When encounter a new dataset / architecture space, how do you propose a new architecture? Do you need to traverse lots of architectures and compare their score? I don't see discussions of this part in the paper.
- Lack of comparison to predictor based methods [2, 3], which also learns an architecture embedding and directly predict the accuracy. It could be better if the authors can show that learning the Gaussian kernel function rivals.
- For the learning process of the architecture / dataset embedding, do you just leverage the previous checkpoints or train them by yourselves? If it does, do you include the training cost into the NAS search cost?
- Another concern is that all the experiments are shown in the NAS-Bench-201 space, which is pretty small. Is it possible to show the result on larger space, e.g., DARTS space. This can also show that the learnt embedding can transfer across spaces.

[1] Yang et al. "OBOE: Collaborative filtering for AutoML model selection." KDD 2019.
[2] Yan et al. "Does Unsupervised Architecture Representation Learning Help Neural Architecture Search?" NeurIPS 2020.
[3] Ning et al. "A Generic Graph-based Neural Architecture Encoding Scheme for Predictor-based NAS." ECCV 2020.

**Summary Of The Paper:**

This paper proposes a transfer NAS approach by leveraging the previous knowledge, i.e., (architecture, dataset, accuracy) triples.
It uses a graph neural network to encode the architecture information, a set transformer to encode the dataset information, and then learning a kernel function with an MLP. Empirical results on NAS-Bench-201 are strong.

**Summary Of The Review:**

The quality of this paper is generally high. However, there are some problems in the method (how to propose new architecture) and the empirical evaluation part (lack of comparison, small search space). I would like see the rebuttal and then adjust my rating.

---

> ### Author Response · Authors · 2022-11-19
> **Initial Response**
>
> Regarding encountering new dataset: first of all, for starting a NAS run on a new dataset, we use the 5 top-performing architectures from the meta-training dataset as a starting point (for TNAS but also for all HPO baselines) (as described in the ‘Results for Hypothesis 1’ paragraph).  After that, in each iteration of TNAS, we need to find the architecture that maximizes the acquisition function in the BO loop. For this, we simply greedily compute the acquisition function on all architectures from the space, as the cost for doing this is negligible in comparison to training time. On search spaces, where this is not easily doable (e.g., for the new experiment on the MobileNet V3 space we added, Figure 7), one can resort to other strategies, e.g., randomly sampling K architectures and choosing the one out of the K that maximizes the acquisition function, or a combination of random sampling and local search [1]. Thank you for making us aware that this information is indeed missing. We added this information in the revised version of our paper (Appendix C).
>
> Regarding lack of comparison to predictor-based methods: the work closest to ours, namely MetaD2A, is in fact a predictor-based method, and we thoroughly compare to that work. But thanks for pointing us towards more predictor-based approaches. We added the results of Yan et al. in Table 3 of the revised paper. Ning et al. unfortunately do not report accuracies on NB201 and we weren’t able to run their method due to time constraints. With the already existing 10 baselines + Yan etl., i.e., a total of 11 baselines, we think we covered a fast number of baselines and many different approaches. We hope you agree.
>
> Regarding the learning process of embeddings, we simply use the meta-training data provided by MetaD2A (available at https://github.com/HayeonLee/MetaD2A ). This is to ensure a fair comparison. The costs for running our meta-training phase is approx. 4 GPU hours (compared to 21 GPU hours for MetaD2A), which is a one-time cost (as the same meta-trained model is used for all test-datasets). We highlight that not just TNAS and MetaD2A benefited from the meta-training stage – as mentioned in the previous point, we also warm-start all HPO baselines from the top-5 architectures from the meta-training set.
>
> Regarding your concerns with NB201 search space: we conducted additional experiments on the MobileNetV3 space (which contains more than 10^19 architectures according to MetaD2A), compared to only 15.6k architectures in NB201. Please refer to Appendix E, Figure 7 for the results. Unfortunately, on this search space, we couldn’t get access to the meta-training dataset that MetaD2A use, so we use the pretrained graph and dataset encoders provided by MetaD2A, and randomly initialize the embedding for the deep kernel. We use 5 randomly sampled architectures from the MobileNetV3 search space as an initial design for the GP. The results show that TNAS outperforms MetaD2A on 3 out of the 4 datasets we tested on, while reaching at least the same final performance than MetaD2A on the remaining benchmark.
>
> [1] White et al., BANANAS: Bayesian Optimization with Neural Architectures for Neural Architecture Search

---

> > ### Comment · Reviewer_3f7e · 2022-11-22
> > **Thanks for the response**
> >
> > Thank you for the response! The rebuttal resolved my concern, and I would like to raise my recommendation.

---

### Official Review · Reviewer_9VTq · 2022-10-23

**Confidence:** 5
**Correctness:** 3
**Technical Novelty And Significance:** 3
**Empirical Novelty And Significance:** 3
**Recommendation:** 8

**Clarity, Quality, Novelty And Reproducibility:**


The quality and clarity are good. The originality is median.



**Details Of Ethics Concerns:**


There is no ethics concern.



**Strength And Weaknesses:**


(Negative) The neural frame transfer technique is not a new technique. Prior arts have had some exploration.

(Positive) This article provides the code and also provides the "NAS Best Practices Checklist." This is very positive because the reproducibility of NAS is widely questioned.


(Positive) The method of this article is very simple.


(Positive) This article introduces Dataset encoding, which is very reasonable.


(Negative) In the article, the definition of architecture encoding is very vague. Please give more details of it.


(Negative) During the acquisition of dataset encoding, positive and negative samples need to be randomly sampled. Is dataset encoding random? If so, please quantify this randomness or expectation.

(Positive) The experimental results in this paper are very good (Figure 2 and Figure 3).


(Negative) It would be better to add experimental validation in real open environments, such as the original ImageNet dataset and the open neural architecture search space. After all, neural architecture transfer and search techniques are meant to be used in real open environments. If the search can only be carried out in these fixed search spaces, its practical significance will be greatly reduced.


(Positive) This article performs a full ablation analysis.


**Summary Of The Paper:**


Summary:
This paper proposes a neural architecture transfer technique. Although this problem is not new, this paper takes some reasonable measures. Its performance is worthy of recognition. This article is an above-average article and deserves a weak acceptance.



**Summary Of The Review:**


See "Summary Of The Paper." This article is an above-average article and deserves a weak acceptance.

---

> ### Author Response · Authors · 2022-11-19
> **Initial Response**
>
> Thank you for your valuable feedback and pointing out the many positive aspects of our work. Below we’ll address the negative aspects you mentioned.
>
> Regarding novelty of neural network transfer, are there any specific papers you are referring to that we did not discuss in our related work section? Of course, our work is not the very first in this direction, but as we tried to make clear in our related work section, prior work typically comes with drawbacks (e.g., they lack of a probabilistic surrogate and the lack of test time adaptation as MetaD2A, they are limited to few-shot learning (few-shot w.r.t. data set size, not number of function evaluations, they only select one architecture out of a handful of candidates,...). Thus, transfer / meta-learning for NAS  is not a solved problem and our work substantially contributes to solving (at least some) current issues. We revisited the introduction of our work to make this more clear.
>
>
> Regarding the vague architecture encoding description: In Section B of the Appendix, we added further clarification on the process of how TNAS generates an encoding from a graph representation of an architecture using the graph encoder, and how it is meta-trained.
>
>
> Regarding learning the dataset encoding: the dataset encoding is optimized on the meta training dataset.  We re-use the meta training dataset provided by the authors of [MetaD2A]( https://github.com/HayeonLee/MetaD2A) to ensure a fair comparison. This meta training dataset consisted of 4230 training sets (aka tasks) derived from ImageNet, as follows [cited from MetaD2A]:
> “We compile ImageNet-1K (Deng et al., 2009) as multiple sub-sets by randomly sampling 20 classes with an average of 26K images for each sub-sets and assign them to each task.[...] We search for the set-specific architecture of each sampled dataset using random search among high-quality architectures which are included top-5000 performance architecture group on ImageNet-16-120 or GDAS (Dong & Yang, 2019b). [...] we additionally collect 2,920 tasks through random sampling.“
> To rephrase, 2920 out of the 4230 tasks consist of randomly sampled data points + randomly sampled architectures (from the NB201 search space) - we assume ‘randomly’ here refers to sampling uniformly at random. The remaining 4230-2920=1310 tasks consistent of random data + top performing architecture (rather than a randomly sampled one).
> We hope this actually answered your question, as we were not sure whether we understood your question correctly. If not, we’d ask you to clarify what you meant so that we can address your question more properly.
>
> Regarding experiments on real open environments: NB201 is a widely used benchmark in NAS research as it makes it easy and cheap to run various experiments due to its tabular nature. In general, cell-based search spaces are the de-facto standard when it comes to search spaces and we feel very few people are working with “open” search space (as they make it really hard to make fair comparisons). Nonetheless, we added an experiment on a MobileNet V3 search space, which we consider much closer to real world environments as many engineers and scientists use versions of MobileNet for real world applications. Please refer to Appendix E, Figure 7 for the results. Unfortunately, on this search space, we couldn’t get access to the meta-training dataset that MetaD2A use, so we use the pretrained graph and dataset encoders provided by MetaD2A, and randomly initialize the embedding for the deep kernel. We use 5 randomly sampled architectures from the MobileNetV3 search space as an initial design for the GP. The results show that TNAS outperforms MetaD2A on 3 out of the 4 datasets we tested on, while reaching at least the same final performance than MetaD2A on the remaining benchmark.
>
> PS: You wrote ‘(Negative) The experimental results in this paper are very good (Figure 2 and Figure 3).’ - we assume this is a typo, as this is something positive?

---

> ### Comment · Reviewer_9VTq · 2022-12-11
> **Responses to the authors' responses:**
>
>
> I really appreciate the great effort the authors have made to respond to my concerns. For example, the authors successfully addressed my concerns in the following areas: novelty, architecture encoding definition, dataset encoding randomness, and open search.
>
> I also read the excellent and valuable comments and suggestions from other reviewers. Congratulations to the authors for being recognized.
>
> In summary, since the authors responded well to my concerns and the article does make sense, I will increase my rating.

---

### Official Review · Reviewer_kATh · 2022-10-24

**Confidence:** 3
**Correctness:** 3
**Technical Novelty And Significance:** 2
**Empirical Novelty And Significance:** 3
**Recommendation:** 6

**Clarity, Quality, Novelty And Reproducibility:**

Overall, some parts of this paper need to be clarified further (see weaknesses above), and the novelty seems to be limited to me.

**Strength And Weaknesses:**

## Strengths
1. Different from (Lee et al., 2019), this paper introduces deep-kernel Gaussian Processes as a probabilistic performance surrogate and intends to adaptively update this surrogate using newly evaluated architectures on the test task.
2. This paper introduces BO to achieve query-efficient (i.e., search-efficient) optimization.


## Weaknesses
1. From my understanding, this paper follows the idea of (Lee et al., 2019) (i.e., meta-learn the representation of datasets and architectures and then predict the performance of architectures based on this learned representation) and only makes a few changes to the method, i.e., replacing its deterministic performance predictor with a probabilistic performance surrogate (i.e., the deep-kernel Gaussian Processes) and using BO for the optimization based on existing works. In this view, this paper mainly combines techniques from different fields for its method and therefore the novelty of this paper is kind of weak to me.
2. In the abstract and introduction section, this paper doesn't provide a clear and detailed demonstration of its motivations, which not only makes me feel confused about why this paper is needed in the literature but also makes the contributions and the importance of this paper less convincing to me. Specifically, this paper only gives a brief introduction to the line of NAS using transferred information across datasets without elaborating more on why these existing works can not well satisfy the demand in practice and why the method in this paper is needed. Moreover, the introduction section of this paper lacks a comparison with its most related work (Lee et al., 2019) to justify the necessity of proposing the method in this paper.
3. Some claims of this paper are not well supported. For example, in the related work section, this paper does not show (a) why the missing trade-off between exploration and exploitation in (Lee et al., 2019) is important and desirable for NAS using transferred information across datasets, and (b) why deep-kernel Gaussian Processes (GP) rather than GPs using the existing kernels are needed in this paper. Moreover, the result of Table 2 in this paper (a) can not unveil the drawbacks of MetaD2A, i.e., the missing exploration vs. exploitation tradeoff and the non-adaptive update in the optimization process and (b) may not support that these drawbacks lead to the stagnated performance of MetaD2A (located at the "Results for Hypothesis 2" of Sec. 5).
4. While this paper claims that the method in this paper can adapt to function evaluations on the test task, Sec. 3 can not well support it because of the missing details of BO. I highly recommend the authors provide the details of how BO works to further justify this point.
5. The empirical results in this paper show that in most cases the proposed method can only marginally improve over MetaD2A. More experiments that can distinguish these two methods may help to justify the superiority of this newly proposed method.

## Questions
1. What's the training cost for w in your proposed method? While there is only marginal improvement over existing NAS from scratch in Figure 2, the total cost (including the training cost for w and the search cost in the x-axis of Figure 2) for the method in this paper may become a problem in practice since it may be more time-consuming.
3. As the w in this paper is trained based on the meta-datasets M, how does this method perform for the NAS problem with out-of-distribution datasets? Will it become a big issue for the proposed method? Because when only considering in-distribution datasets, directly transferring the selected architectures from other datasets can already provide competitive performances with compelling search costs as shown in Figure 2. So, the proposed method in this paper may not be necessary for this scenario. Meanwhile, if the proposed method can not guarantee its performance for the out-of-distribution datasets, this may also hinder the application of this method in practice.


**Summary Of The Paper:**

As most NAS algorithms search for well-performing architectures from scratch given the target dataset, this paper follows the existing line of transferring information across different datasets to accelerate the search process. Specifically, this paper follows (Lee et al., 2021) to learn the representation of datasets and architectures from many other datasets (i.e., as training data) and then propose to leverage Bayesian Optimization (BO) with deep-kernel Gaussian Processes to find the optimal architecture for a specific dataset. Empirical results show that this method can improve NAS from scratch and also the method in (Lee et al., 2019).

**Summary Of The Review:**

In general, my major concerns lie in the novelty and clarity of this paper. I hope the authors can address my concerns during the rebuttal period.

---

> ### Author Response · Authors · 2022-11-19
> **Thank you for your elaborate review. (part 1)**
>
> Regarding clear and detailed motivation: our motivation is twofold: firstly, from a methods-point-of-view, current NAS approaches typically either suffer from (i) being inefficient (e.g., BBO methods), or (ii) they do not work robustly across datasets (e.g., one-shot methods). This we also find in our experiments. Thus, TNAS aims at solving this issue by being as efficient as, e.g., one-shot methods, while being as robust as more classical BBO approaches. Secondly, TNAS is very much motivated by what human engineers and scientists do when solving new deep learning problems – they don’t start from scratch when looking for a suitable architecture. Rather, they leverage their experience and warmstart the new problem from that experience. So, our work is naturally motivated by the fact that TNAS is mimicking what human experts do. We are sorry if this did not become clear in the introduction, we revised the introduction and also added a table, Table 2 (Appendix D), for a direct comparison with MetaD2A. In general, a more thorough discussion can be found in the related work section rather than the introduction.
>
> Regarding the claims of our paper: (a) Figure 3 shows that MetaD2A on almost all data sets very quickly stagnates (sometimes with good, sometimes with bad performance), indicating the lack of exploration and the lack to adapt to the new datasets, which is necessary to improve upon the current best solution. In contrast, for TNAS one can see consistent improvements over time (for most datasets), showing that novel solutions are discovered. We also added Figure 8 (Appenidx E) for a closer look, which shows all the proposed architectures. While there is no learning curve at test-time for MetaD2A, we can see that TNAS picks up on information gathered at test-time and over the course of evaluation tends to propose better performing architectures, which is not the case for MetaD2A.
> (b) Why deep-kernel GPs and not vanilla kernels: there are 2 main reasons for using the deep-kernel GP. 1) there is simply no off-the-shelf kernel that can deal with (architecture, dataset) tuples as an input. This is a very specialized setting and vanilla kernels such as Matern expect real-valued inputs, so it is technically not possible. 2) Vanilla kernels have very few learnable parameters, thus they can not leverage the meta training data. In contrast, TNAS employs deep learning (for the dataset encoding, for the architecture encoding, for processing these encodings) to leverage the strength of deep learning to benefit from vast amounts of data during meta training.
> (c) Table 2. Table 2 (Table 3 in the current version of the paper) shows that TNAS statistically significantly outperforms MetaD2A on four out of six datasets, while being on-par on the remaining two datasets. Furthermore, Figure 3 (rather than Table 3) shows the drawbacks of MetaD2A, as already elaborated above (in short, MetaD2A quickly stagnates, which indicates the lack of exploration and test time adaptation).
>
> Regarding the adaptation to function evaluations at test tasks: as explained in Section 3.1, the posterior of the GP providing estimates on new data points is given by a Gaussian distribution with mean and covariance matrix as in equations (1), (2). New function evaluations on a novel test task will be considered as part of the covariance matrix in these equations, thus impacting the prediction on new data points. We added a paragraph at the end of the methods section (3.3) to clarify.
>
>  Costs for optimizing meta parameters w: the costs for the meta-training stage is approx.. 4 GPU hours.  We simply re-use the meta-training dataset provided by [MetaD2A]( https://github.com/HayeonLee/MetaD2A) . Please note, this is a  **one-time cost** you need to invest upfront (as the same meta-trained model is used for all test-datasets). Also, we highlight that not just TNAS and MetaD2A benefited from the meta-training dataset – as mentioned in “Results for Hypothesis 1” paragraph, we warm-start all HPO baselines from the top-5 architectures from the meta-training set. Thus, these baselines also benefit from the meta training dataset.

---

> > ### Author Response · Authors · 2022-11-19
> > **part 2**
> >
> > Regarding empirical results: please note that, due to the nature of the benchmark, very often differences are small (as can also be seen in MetaD2A). Also, note that the intention of TNAS is **not** to provide SOTA performance on a specific benchmark by designing and tuning an approach for a specific benchmark (as is often the case in NAS research). Rather, the intention of TNAS is to work **robustly across benchmarks**, with consistent strong results. This is clearly visible in Figure 4, where we show that, across all benchmarks and across different compute budgets, TNAS always ranks 1st or 2nd among all baselines. In contrast, all baselines have a higher variance in their ranking. E.g., while MetaD2A performs very strong on Cifar10, Cifar100 and AIRCRAFT, it performs poorly on MNIST, PETS and SVHN. DrNAS performs very strong on Cifar100, while being ranked last on MNIST. I.e., the consistency we achieve with TNAS does not hold for the baselines considered.
> >
> > Nonetheless, we updated the paper with additional experimental results on the MobileNet V3 space, which is more realistic and much larger compared to NB201. Please refer to Appendix E, Figure 7 for the results. Unfortunately, on this search space, we couldn’t get access to the meta-training dataset that MetaD2A use, so we use the pretrained graph and dataset encoders provided by MetaD2A, and randomly initialize the embedding for the deep kernel. We use 5 randomly sampled architectures from the MobileNetV3 search space as an initial design for the GP. The results show that TNAS outperforms MetaD2A on 3 out of the 4 datasets we tested on, while reaching at least the same final performance than MetaD2A on the remaining benchmark.
> >
> > Regarding performance on out of distribution datasets: first of all, a key assumption for any meta or transfer learning approach is, that the target dataset shares similarities with the meta training data. If there are no similarities, there is simply nothing to meta-learn. Rather than being a limitation of TNAS, this is a limitation of the general idea of transfer/meta learning. Secondly, we’d argue that some of the datasets we consider are in fact, to some extend, out of distribution – our meta training data set (ImageNet) consists of natural images across various categories. We transfer from that to fairly different kind of data distributions: (i) pictures of house numbers, typically showing printed numbers on a  monotonous background (SVHN), (ii) non-natural black and white image with digits (MNIST) and (iii) a classification problem very much specialized on a single object category, namely airplanes (AIRCRAFT).

---

> > > ### Comment · Reviewer_kATh · 2022-11-24
> > > **Thank you for the response**
> > >
> > > I thank the authors for the detailed response. I hope the authors can address my concern about the novelty of this paper (i.e., weakness 1) and also include all these discussions in the revised paper. In addition, for the response to weakness 3, without additional ablation study, I do not think the authors can directly contribute the improvement to the exploration nature of the proposed method. Moreover, I hope that the authors can compare the proposed method with directly transferring the searched architectures into the target task to further clarify the advantages of introducing meta-learning into NAS especially when the tasks share small similarity (this can help people better understand in what kind of scenario the proposed method should be used). Because the proposed methods will introduce additional search cost compared with direct transferring the searched architectures.

---

> > > > ### Author Response · Authors · 2022-11-25
> > > > **Additional clarifications**
> > > >
> > > > In the camera ready, we will gladly clarify the aspects you mention in terms of novelty, by delineating from Lee et al., 2019 in the introduction.
> > > >
> > > > Regarding the exploration nature of Bayesian optimization (BO), there are two arguments that can be brought to attention.
> > > >
> > > > First of all, BO is well-known to possess an exploration aspect due to using the posterior variance of the probabilistic surrogate (our deep-kernel GP) inside the acquisition function. An illustration of the BO mechanism with GPs is offered in Figures 1-2 & 6 in [a]. The exploration part (as clarified in the example of the referenced document) relies on having a probabilistic surrogate, which is a crucial ingredient the baseline (Lee et al., 2021) is lacking.
> > > >
> > > > Secondly, during the rebuttal, we actually added a new ablation in Figure 8 that shows the progression of the accuracy of architectures during the search results. We clearly see that in comparison to the non-exploratory baseline, our method improves the quality of the results vs. time. The finding is in line with the expected behavior of Bayesian optimization. Unfortunately, our latent space where BO is conducted has dozens of dimensions (architecture representation + dataset encoding, details Appendix A and B), and is not possible to plot the loss surface in order to visualize the results as in the 1-d example of Figures 1-2 & 6 in [a].
> > > >
> > > > Regarding the effect of meta-learning vs. transferring the top architectures, we already presented that analysis in Section 5.1, Figure 6. In the rightmost column, you find the difference between a meta-learned surrogate, vs. no meta-learned surrogate when both of them use the top-5 architectures of the source task (top 5 performers in ImageNet) as the initial observations for the Bayesian optimization. The gap between the blue and green lines shows the gain of meta-learning. This analysis also shows the performance if you do not transfer the top architectures to the target task, and instead use random initial architectures (red and orange lines).
> > > >
> > > > In addition, all the baselines in Figure 2 (Random Search, GP, etc. ) are also warm-started with the top-5 configurations from the meta-training dataset, as clarified in Section 5, Results for Hypothesis 1, and also mentioned again in "part 1" of our response. A meta-learned surrogate with explorative Bayesian optimization outperforms non-meta-learned search variants that transfer the top configurations from the source dataset. A meta-learned surrogate with explorative Bayesian optimization outperforms non-meta-learned search variants that transfer the top configurations from the source dataset.
> > > >
> > > > Therefore, your intuition is correct. Transferring architectures to the target task from a similar source task (e.g. Imagenet to Cifar10) helps the search, compared to a random set of initial architectures. However, the meta-learning of the surrogate (a.k.a. performance predictor) gives a clear lift on top of simply transferring good architectures.
> > > >
> > > > [a] Brochu et al., 2010 (https://arxiv.org/pdf/1012.2599.pdf)

---

> > > > > ### Comment · Reviewer_kATh · 2022-11-26
> > > > > **Thank you for the additional clarifications**
> > > > >
> > > > > I thank the author(s) for the additional responses. Now, most of my concerns have been addressed. So, I decided to increase my score.

---

### Official Review · Reviewer_AX7F · 2022-10-26

**Confidence:** 4
**Correctness:** 3
**Technical Novelty And Significance:** 4
**Empirical Novelty And Significance:** 2
**Recommendation:** 8

**Clarity, Quality, Novelty And Reproducibility:**

I think clarity of this paper is needed to be improved as there are mixing between their contributions and previous works' contributions.
Actually, applying BO to NAS is not new, yet, to my knowledge, this work is the first to use BO for meta-learning-based predictor for NAS and addresses the important problem in meta-learning-based NAS.
On the NAS-Bench-201 benchmark and multiple unseen datasets, their experiments are solid to support this work. Thus, i think the overall quality is good, yet, one caveat is that NAS-Bench-201 is rather small and narrow benchmark.



**Strength And Weaknesses:**

- Strengths
   - I think this work well addressed the main limitation of MetaD2A which is the meta-learning-based transferrable NAS method. As this paper said, the meta-learning-based predictor proposed by MetaD2A can be exploited to unseen datasets after once meta-training, which significantly reduces the search time for unseen datasets. However, MetaD2A only exploits the meta-learned predictor, can not adapt to the unseen dataset even if the unseen task provides few-shot samples. This work combines the BO method to tackle the problem of transferrable predictors by allowing them to reflect feedback from unseen tasks.

- Weaknesses
  - I think while the contribution of this work in the structure of predictor is limited, this work assigned the part of the paper for them too much. For example, using both transformer-based set encoder and graph encoder together is almost same with MetaD2A, this paper described it  as the figure and performed ablation study about that. I think it would be better to emphasis the difference between MetaD2A and this work or BO + predictor as a Figure.

**Summary Of The Paper:**

This work proposes a transferrable surrogate for NAS based on Bayesian Optimization with deep-kernel Gaussian Processes. The proposed predictor can be adapt to unseen datasets rapidly by significantly reducing the search cost of NAS. On the NAS-Bench-201 and multiple unseen datasets, this work outperformed recent NAS methods including MetaD2A for the performance of the obtained architecture and search efficiency.

**Summary Of The Review:**

The good points are that this work successfully addressed the important problem of the meta-learning-based prediction model in NAS with solid experiments. The bad points are that BO is not new in NAS domain, the composition of the paper is needed to be improved, and the benchmark that they used is small.

---

> ### Author Response · Authors · 2022-11-19
> **Thanks for your feedback.**
>
> Thanks very much for your feedback and pointing out the key advantages of our method compared to MetaD2A.
>
> We revised the paper according to your suggestions (mention more explicitly that we borrow the architecture and dataset encoder from MetaD2A + better comparison of MetaD2A and TNAS); in particular, we added Table 2, comparing both approaches. We’d appreciate further feedback here in case you don’t find our changes don’t address your concerns.
>
> Regarding your concern that BO has been used in the NAS domain before: indeed, vanilla BO has been used in NAS before. However, we are the first (to the best of our knowledge) to extend BO for NAS to the transfer/meta-learning setting, by being the first that leverage transfer HPO methods for NAS. We think employing transfer/meta-learning for NAS is an important next step for NAS as 1) it combines the best of both worlds (being as robust as BBO methods, while being as efficient as one-shot methods), thus making NAS practical and 2) our idea is very natural as our method mimics what human engineers and scientists do - they learn from experience and don’t start from scratch.
>
> Lastly, to address your concerns about the NB201 search space, we ran TNAS on the MobileNet V3 search space (which is considerably larger and closer to real-world applications). Please refer to Appendix E, Figure 7 for the results. Unfortunately, on this search space, we couldn’t get access to the meta-training dataset that MetaD2A use, so we use the pretrained graph and dataset encoders provided by MetaD2A, and randomly initialize the embedding for the deep kernel. We use 5 randomly sampled architectures from the MobileNetV3 search space as an initial design for the GP. The results show that TNAS outperforms MetaD2A on 3 out of the 4 datasets we tested on, while reaching at least the same final performance than MetaD2A on the remaining benchmark.

---

> > ### Comment · Reviewer_AX7F · 2022-11-27
> > **Thank you for the response.**
> >
> > The authors addressed my main concerns by 1) providing the comparison Table to emphasize their contributions compared with the baseline method (MetaD2A) and 2) providing results on larger search space - MobileNetV3 search space.
> >
> > Minor issues
> > 1) I think that moving Table 2 and Figure 7 to the main paper would be great.
> > 2) Concept figure is too simple. Providing more details for your method would be great.
> >
> > I raised my score from 6 to 8.

---

### Author Response · Authors · 2022-11-19
**General Response**

We thank all the reviewers for their elaborate and constructive feedback.
We made the following changes to our paper:
-  We rephrased some paragraphs in the introduction and related work according to suggestions by the reviewers.
-  We added details on the Bayesian Optimization part of TNAS (how to get new candidate architectures, how to adapt at test time), please refer to Section 3.3. and  Appendix C., as reviewers rightfully asked for these details.
-  We added a table comparing the key ingredients of TNAS and MetaD2A (Table 2 in Appendix D), as this was also a common request.
-  We added additional experimental results on the MobileNet V3 space, which is considerably larger than NB201. Due to time constraints, we used the pretrained weights of a [Once-for-all](https://github.com/mit-han-lab/once-for-all.git) MobileNet V3 supernet, and only finetuned the suggested architectures for 1 epoch on the test dataset. The results are shown in Figure 7 (Appendix E), as this was also a common request. On this space, TNAS outperforms MetaD2A on 3 out of 4 datasets, while reaching at least the same final performance than MetaD2A on the remaining benchmark.

Most of the changes are right now in the Appendix for pointing them out more clearly to you. In case our work will get accepted, we plan to merge most of them into the main paper.
We hope that with these changes we addressed all major concerns.

---

### Decision · Program_Chairs · 2023-01-20

**Decision:**

Accept: notable-top-5%

**Justification For Why Not Higher Score:**

Highest already

**Justification For Why Not Lower Score:**

Three out of four reviewers were excited about the paper.

**Metareview: Summary, Strengths And Weaknesses:**

All reviewers were positive about the paper. Hence, the decision is to recommend the paper for acceptance. The authors are encouraged to make the necessary changes to the paper to the best of their ability following the reviewers' comments.


**Note From Pc:**

if the above contains the word "oral" or "spotlight" please see: "oral" presentation means -> notable-top-5% and "spotlight" means -> notable-top-25%. As stated in our emails, we are disassociating presentation type from AC recommendations